# Concurrent use of low complexity automated NAATs for TB diagnosis and detection of resistance: A cost-effectiveness analysis

Suvesh Kumar Shrestha[1], Funeka Bango[2], Pushpita Samina[3], Alexei Korobitsyn[4], Nazir Ismail[4,5], Alice Zwerling[1]*

1 School of Epidemiology & Public Health, University of Ottawa, Ottawa, Canada, 2 South African Medical Research Council, Cape Town, South Africa, 3 McMaster University, Hamilton, Canada, 4 Global Tuberculosis Program, World Health Organization, Geneva, Switzerland, 5 Department of Clinical Microbiology and Infectious Diseases, Faculty of Health Sciences, Wits University, Johannesburg, South Africa

* azwerlin@uottawa.ca

## Abstract

Current TB diagnostics relying on respiratory samples are less effective in children (under 10 years), children living with HIV (CLHIV) and adult people living with HIV (PLHIV). Concurrent testing using low-complexity automated nucleic acid amplification tests (LC-aNAATs) on multiple sample types and Lateral Flow Lipoarabinomannan (LF-LAM) improves diagnostic performance, but concerns about cost-effectiveness remain. To inform WHO guideline development group (GDG) recommendation we developed stochastic decision analysis model to evaluate the cost-effectiveness of concurrent LC-aNAAT testing in these populations in Malawi and the Philippines. The analysis, conducted from a healthcare system perspective estimated incremental cost per disability-adjusted life year (DALY) averted using country-specific parameters from published literature and systematic review conducted for GDG. Concurrent testing was assessed using LC-aNAAT on respiratory and stool samples for children, LC-aNAAT on respiratory samples and LF-LAM for PLHIV, and a combination of both approaches for CLHIV. Concurrent testing in children compared to respiratory samples had incremental cost per DALY averted of $253 (95% uncertainty interval (UI): $123 to $2317) in Malawi and $156 (95% UI: $79 to $888) in the Philippines. For PLHIV, concurrent testing had an incremental cost per DALY averted of $42 (95% UI: $18 to $345) in Malawi and $28 (95% UI: $12 to $249) in the Philippines. In CLHIV, concurrent testing had an incremental cost per DALY averted of $43 (95% UI: $28 to $89) in Malawi and $29 (95% UI: $18 to $63) in the Philippines. Sensitivity analyses highlighted TB prevalence and respiratory sample availability as key influencers of cost-effectiveness. In scenarios with higher TB prevalence, cost-effectiveness improved, while an increase in the probability of producing respiratory sample resulted in worsening cost-effectiveness estimates. Concurrent use of LC-aNAATs on multiple sample types and

**Data availability statement:** This study is a modeling analysis based on data extracted from publicly available sources, including previously published literature. All relevant data and parameter values used in the model are provided in the Supporting information files.

**Funding:** This work was supported by the World Health Organization to Alice Zwerling. The funders had no role in study design, data collection and analysis, decision to publish, or preparation of the manuscript.

**Competing interests:** The authors have declared that no competing interests exist.

in conjunction with LF-LAM emerged as highly cost-effective strategy for diagnosing TB among children, PLHIV, and CLHIV.

## Introduction

Tuberculosis (TB) remains a major global health issue and a leading cause of mortality. The COVID-19 pandemic has significantly disrupted TB diagnostic services, leading to an 18% reduction in newly diagnosed cases in 2020 compared to 2019 [1]. This gap in case detection, particularly pronounced among children and people living with HIV (PLHIV) [2–4], is largely due to challenges with respiratory specimen collection and the paucibacillary nature of TB in these groups, coupled with a lack of highly sensitive point-of-care tests [5].

The World Health Organization (WHO) End TB Strategy underscores the need for rapid diagnostics and universal access to quality-assured drug susceptibility testing (DST) [6]. WHO's 2021 guidelines recommend using molecular rapid diagnostics (mWRDs) such as Xpert MTB/RIF, Xpert MTB/RIF Ultra, and Truenat as initial tests for TB diagnosis and drug resistance detection. These low-complexity automated nucleic acid amplification tests (LC-aNAATs) require minimal infrastructure and basic laboratory skills [7]. Tuberculosis diagnosis relies heavily on evaluation of sputum and LC-aNAATs on respiratory samples [8]. However, among children, PLHIV and children living with HIV (CLHIV) who are unable to produce sputum, this approach may lead to low sensitivity and poor diagnostic yield.

Concurrent testing on multiple specimens has been shown to improve diagnostic yield to diagnose TB [9]. Some studies have shown that urine sample LF-LAM used concurrently with respiratory LC-aNAAT may be complementary for the diagnosis of tuberculosis among PLHIV [10,11]. Studies done on children, have found that collecting different respiratory sample types (e.g., induced sputum and nasopharyngeal aspirate) may be beneficial [12,13]. Given the low sensitivity of respiratory samples for children, PLHIV and CLHIV, there is an urgent need for non-sputum-based testing utilizing alternative sample types that could be performed concurrently with respiratory samples to increase diagnostic yield.

While concurrent testing using multiple sample types offers the potential to increase diagnostic yield in key populations, the additional costs and cost-effectiveness of these approaches are unknown. Concurrent testing may pose additional upfront testing costs, and full economic evaluations are crucial to understand the cost-effectiveness of these approaches and to inform decision-making given the limited resources faced globally by TB programs. Existing cost-effectiveness studies on concurrent testing are limited to PLHIV but suggest that combining Xpert MTB/RIF with LF-LAM can improve life expectancy and be cost-effective [14–18]. However, there is a gap in evidence regarding concurrent testing of LC-aNAATs in children, PLHIV and CLHIV.

This study aimed to generate economic evidence to support the WHO Guideline Development Group (GDG) in formulating guidance on the concurrent use of LC-aNAATs for the diagnosis of TB among children without HIV, PLHIV, and CLHIV.

The primary objective of this study is to assess the cost-effectiveness of concurrent use of LC-aNAATs on respiratory and stool samples for detecting TB among children, LC-aNAAT on respiratory sample and LF LAM on urine to detect TB in PLHIV, and LC-aNAATs on respiratory and stool samples and LF LAM on urine to detect TB in CLHIV in Malawi and the Philippines.

## Methods

### Study design

We used a decision analysis model to evaluate the cost-effectiveness of concurrent use of LC-aNAAT among three target populations: children without HIV under 10 years, adult PLHIV, and CLHIV under 10 years. The children in this study are under 10 years old as per the WHO definition of Children, and adults are 15 years and above [19]. The analysis was conducted from a healthcare system perspective, spanning a lifetime to capture the long-term impact of TB diagnosis interventions. The models were parametrized and structured to reflect high TB burden settings in sub-Saharan Africa and Asia, using Malawi and the Philippines as emblematic examples. Malawi represents a setting with a high TB and HIV burden [20], with approximately 7.9% of adults living with HIV [21], and a substantial proportion of TB cases occur among PLHIV. In contrast, the Philippines is characterized by a high TB burden but low HIV prevalence (0.2–0.3% among adults) [22], offering a distinct epidemiological context. In this study, LC-aNAAT is assumed to be available at the level of the health system where there is capacity for TB screening and conducting diagnostic tests, consistent with decentralized implementation strategies. However, we did not assume universal availability across all levels of care, recognizing that access to these tools may vary by setting and programmatic capacity. This study is a secondary data modelling analysis based entirely on data extracted from publicly available sources, including published literature and global reports. As no primary data collection or human subjects were involved, ethical approval was not required.

### Intervention, and comparator

For children under 10 years, the intervention compared concurrent use of LC-aNAATs on respiratory and stool samples with LC-aNAAT on respiratory samples alone. Among PLHIV adults, the study compared concurrent LC-aNAAT on respiratory samples and LF-LAM on urine to LC-aNAAT on respiratory samples alone. For CLHIV under 10 years, the intervention involved concurrent LC-aNAATs on respiratory and stool samples alongside LF-LAM on urine, compared to LC-aNAAT on respiratory samples alone.

### Modelling

A decision analysis model developed to assess the cost-effectiveness (CE) of concurrent LC-aNAAT use among children (Fig 1), PLHIV (S1 Fig), and CLHIV (S2 Fig). The model simulated a cohort of presumptive TB cases through diagnostic and treatment pathways, comparing the intervention which concurrent testing with LC-aNAAT in children, PLHIV and CHIL, versus respiratory sample testing alone. The model accounted for variations in specimen availability across subgroups (e.g., use of stool or LF-LAM when respiratory samples were not obtainable). Both arms allowed for clinical diagnosis in cases not confirmed through testing. Bacteriologically confirmed cases underwent rifampicin DST and were initiated on appropriate TB treatment. The model followed all individuals over time, including those with false-negative or false-positive results, to capture unnecessary treatment and additional mortality due to missed diagnoses.

### Assumptions

Several assumptions were made across all models. One assumption was around sample provision where ideally, individuals in intervention arm provides two or more samples including a respiratory sample. For those individuals who were not able to provide a respiratory sample it was assumed to provide at least one alternative sample - stool samples for

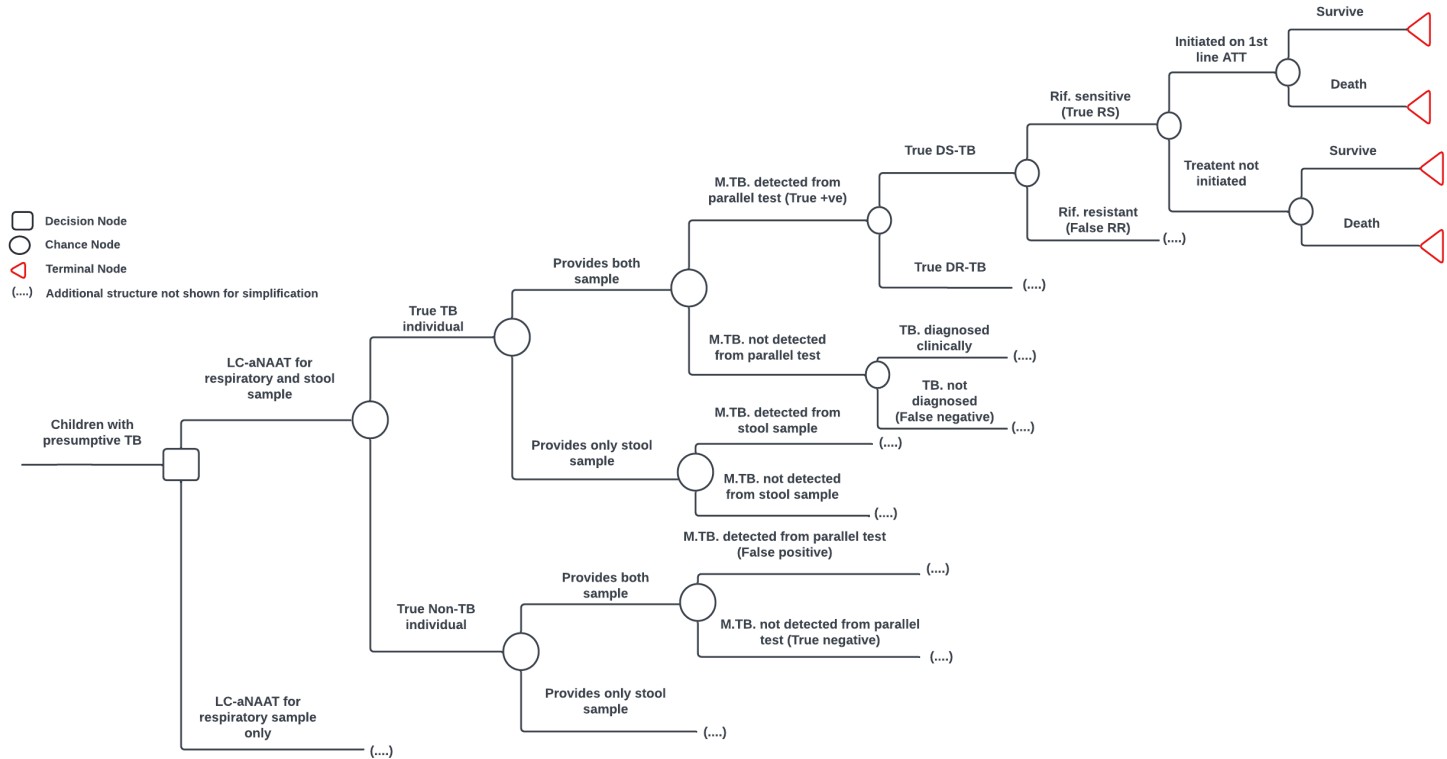

**Fig 1. Simplified model decision structure on the concurrent use of LC-aNAAT in respiratory and stool sample vs. single use of LC-aNAAT in respiratory sample among children with presumptive TB.** Schematically these strategies are separated by a square representing decision node. The circle represents chance nodes where individuals may experience one of several possible events shown on subsequent lines. Dotted lines represent model structure omitted for simplicity. The triangle symbol represents terminal node. may experience one of several possible events shown on subsequent lines. TB = Tuberculosis; LC-aNAAT: Low complexity automated nucleic acid amplification tests; DS: Drugs susceptible; DR: Drugs resistant; Rif: Rifampicin; RS: Rif. Sensitive; RR: Rif. Resistant.

children, urine sample for PLHIV and both stool and urine samples for CLHIV. Additionally, all PLHIV and CLHIV were assumed to be receiving anti-retroviral treatment (ART).

Further, DST, after detecting rifampicin resistance (RR), was not performed in the model; assuming that the resistance was only for rifampicin. It was also assumed that treatment would be initiated for all individuals diagnosed with TB. Those diagnosed with DS TB or clinically diagnosed would be initiated on first-line TB drugs with two months of Isoniazid, Rifampicin, Pyrazinamide and Ethambutol (HRZE) followed by 4 months of HR. Similarly, those diagnosed with RR TB would be initiated on the Bedaquiline, Pretomanid, Linezolid, and Moxifloxacin (BPaLM) regimen.

### Epidemiological, diagnostic accuracy and utility parameters

Epidemiological parameters like country-specific prevalence of TB and drug-resistant TB were taken from published literature and the WHO country profile (S1 Table) [23]. Data concerning the sensitivity and specificity of diagnostic approaches were sourced from an ongoing systematic review of the diagnostic accuracy of LC-aNAAT (S2 Table). Utility data on DALYs were based on previously published literature from relevant countries (S3 Table) [24].

### Cost parameters

Per unit test costs of LC-aNAAT were derived from an ongoing systematic review on cost-effectiveness of LC-aNAAT done in preparation for the GDG meeting and from available published data. The costs of the DS and DR TB treatment

were extracted from different published literature, including a recently published cost-effectiveness study done by Sweeney et. al. on short oral treatment regimens in the countries selected for this study [25] and from the Value TB database [26]. Per unit test costs of other diagnostics were taken from published literature and the Value TB studies, which provide comprehensive information on cost of various tuberculosis interventions. All costs were presented in US dollars and converted to 2024 US using the inflation rate available through US Bureau of Labour Statistics. The S4 Table shows key cost parameters.

### Analysis

This study measured outcomes in terms of total costs, and Disability Adjusted Life Years (DALYs). The primary outcome is the incremental cost per disability-adjusted life year (DALY) averted. Secondary outcomes included incremental cost per additional TB case diagnosed and incremental cost per death averted.

### Uncertainty, sensitivity and scenario analysis

To address parameter uncertainty, we used probabilistic sensitivity analysis with 10,000 Monte Carlo sampling to generate 95% uncertainty interval (UI). One-way sensitivity analysis assessed key inputs' impact on ICER results, presented via tornado diagrams.

To test the impact of various scenarios on cost-effectiveness, scenario analyses were performed. We examined scenarios where the proportion of individuals able to produce respiratory samples varied at 20%, 50%, and 80%. These values were selected to reflect a possible range observed in programmatic settings, where the ability to obtain respiratory samples varies, particularly among children and individuals such as PLHIV and CLHIV. We also explored the scenario where the sensitivity of clinical diagnosis ranges from 20% to 80%, representing a plausible range of diagnostic performance to explore best- and worst-case scenarios under programmatic conditions. Additionally, we considered scenarios with alternate comparators. For children and CLHIV, the alternative comparator was LC-aNAAT on stool samples alone, while for PLHIV, it was the use of LF-LAM alone.

### Ethical statement

This was a hypothetical modeling study that utilized secondary data from previously published sources and did not involve human participants. Therefore, ethical approval and formal consent were not required.

## Results

### Concurrent use of LC-aNAATs on respiratory and stool samples among children with signs and symptoms of TB

When using the emblematic setting of Malawi to parametrize the model, cost-effectiveness modelling found that the use of LC-aNAAT on respiratory samples resulted in an average cost of $144, with a corresponding average 0.93 DALYs. In contrast, the concurrent use of LC-aNAAT on respiratory and stool samples yielded an average cost of $204, and 0.57 DALYs resulting in an incremental cost per DALY averted of $253 (95% UI): $123-$2317) (Table 1).

Similarly, in the Philippines, the cost of LC-aNAAT on respiratory samples is $84, associated with a DALY value of 1.04. Concurrent testing in the Philippines resulted in an average cost of $149, and 0.62 DALYs. With an ICER of $156 per DALY averted, (95% UI: $79 to $888) (Table 1).

In Fig 2A and 2B, the ICER scatterplot based on 10,000 Monte Carlo simulations presents the cost-effectiveness results comparing concurrent use of LC-aNAATs on respiratory and stool samples compared with standalone use on a respiratory sample among children in Malawi and the Philippines, respectively. Results consistently show incremental effectiveness greater than zero, indicating that concurrent testing methods utilizing LC-aNAAT result in improved diagnostic yield, and thus more DALYs averted compared to respiratory sample testing alone.

**Table 1. CEA on concurrent use of LC_aNAAT among children, PLHIV and CLHIV.**

| Target group | Country | Strategy | Cost (US$) | Effectiveness | ICER with 95% UI (US$) |
|---|---|---|---|---|---|
| Children | Malawi | LC-aNAAT on respiratory sample | $114 | 0.93 | Ref |
| | | LC-aNAAT on respiratory and stool sample | $204 | 0.57 | $253 ($123-$2317) |
| | Philippines | LC-aNAAT on respiratory sample | $84 | 1.04 | Ref |
| | | LC-aNAAT on respiratory and stool sample | $149 | 0.62 | $156 ($79-$888) |
| PLHIV | Malawi | LC-aNAAT on respiratory sample | $276 | 2.44 | Ref |
| | | LC-aNAAT on respiratory and LF-LAM | $298 | 1.93 | $42 ($18-$345) |
| | Philippines | LC-aNAAT on respiratory sample | $220 | 2.78 | Ref |
| | | LC-aNAAT on respiratory and LF-LAM | $238 | 2.13 | $28 ($12-$249) |
| CLHIV | Malawi | LC-aNAAT on respiratory sample | $320 | 5.08 | Ref |
| | | LC-aNAAT on respiratory, stool and LF-LAM | $460 | 1.8 | $43 ($28-$89) |
| | Philippines | LC-aNAAT on respiratory sample | $249 | 5.13 | Ref |
| | | LC-aNAAT on respiratory, stool and LF-LAM | $345 | 1.77 | $29 ($18-$63) |

CEA: Cost Effectiveness Analysis; LC_aNAAT: Low complexity automated nucleic acid amplification tests; DALY: Disability Adjusted Life Year, ICER: Incremental Cost-Effectiveness Ratio; US$: United States Dollars, UI: Uncertainty Interval; PLHIV: People Living with HIV; CLHIV: Children Living with HIV.

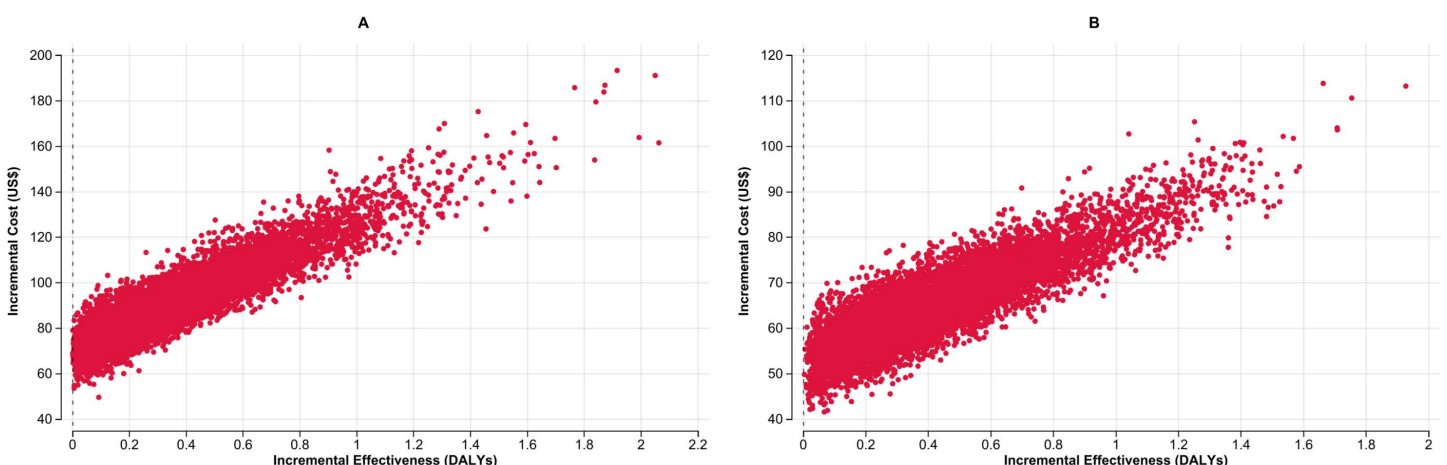

**Fig 2. Incremental cost-effectiveness scatter plots for the concurrent use of LC-aNAATs among children in Malawi (Panel A) and the Philippines (Panel B): Each scatterplot compares the ICER of concurrent testing versus respiratory sample testing alone, showing incremental cost (X-axis) against incremental effectiveness in DALYs averted (Y-axis).** Red dots represent 10,000 probabilistic simulations. LC-aNAAT: Low-complexity automated nucleic acid amplification test; DALY: Disability-Adjusted Life Year; ICER: Incremental Cost-Effectiveness Ratio; US$: United States Dollars.

The CE acceptability curve analysis, shown in Fig 3A and 3B for Malawi and the Philippines, respectively, illustrates the proportion of simulations where LC-aNAAT used in concurrent testing among children is deemed cost-effective compared to testing respiratory sample alone, at varying WTP thresholds. In Malawi, at a WTP threshold of $300, concurrent testing is preferred in over 50% of simulations among children. Similarly, in the Philippines, at a WTP threshold of $180, concurrent testing is preferred in over 50% of simulations among children.

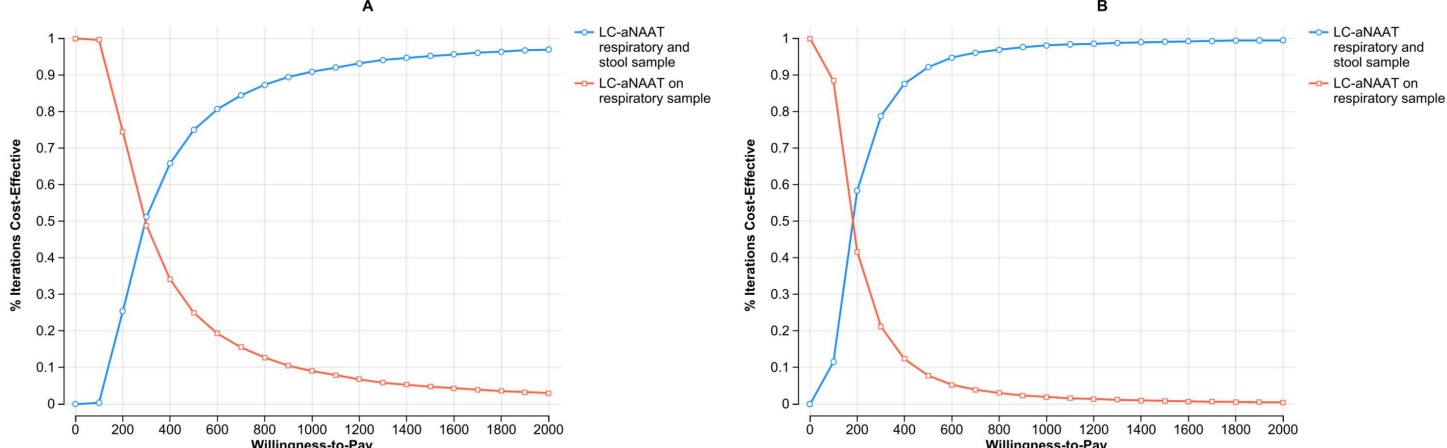

**Fig 3. Cost effectiveness acceptability curve on concurrent use of LC-aNAATs on children in Malawi (Panel A) and the Philippines (Panel B): CE acceptability curve: Represents the proportion of simulations where LC-aNAAT in concurrent testing is considered cost-effective compared to respiratory sample testing among children, at varying WTP thresholds.** Blue circles depict LC-aNAAT in concurrent testing; red circle LC-aNAAT in respiratory sample testing. LC-aNAAT: Low complexity automated nucleic acid amplification tests; CE: Cost Effectiveness; WTP: Willingness to Pay.

In Table 2 we observe the results of ICER per additional TB case diagnosed and ICER per death averted. The ICER per additional case diagnosed was $2,241 (95% UI: $1573-$5240) in Malawi and $1518 (95% UI: $1053-$3371) in Philippines. Similarly, the ICER per death averted was $14,594 (95% UI: $7,185-$128,002) in Malawi and $9965 (95% UI: $5000-$56896) in the Philippines. In Malawi, concurrent testing detected 40 additional TB cases and averted 6 deaths, while in the Philippines, it detected 43 additional TB cases and averted 7 deaths per 1,000 children compared to the comparator (S5 Table).

**Concurrent use of LC-aNAAT on respiratory sample and LF-LAM on urine among presumptive TB adult PLHIV**

The study finding presented in Table 1 shows the cost effectiveness of concurrent use of LC-aNAAT with LF-LAM among PLHIV when using the emblematic setting of Malawi and the Philippines to parametrize the model respectively. In Malawi, the cost of implementing LC-aNAAT on respiratory sample amounts to an average cost of $276, with a corresponding

**Table 2. ICER per additional TB case diagnosed and ICER per death averted.**

| Target group | Country | ICER per additional TB case diagnosed | ICER per death averted |
|---|---|---|---|
| Children | Malawi | $2241 ($1573-$5240) | $14594 ($7185-$128002) |
| | Philippines | $1518 ($1053-$3371) | $9965 ($5000-$56896) |
| PLHIV | Malawi | 340 (272- 435) | $1174 ($504-$10323) |
| | Philippines | $290 ($233-$373) | $949 ($412-$8329) |
| CLHIV | Malawi | $1026 ($870-$1239) | $2425 ($1600- $4911) |
| | Philippines | $701 ($612- $842) | $1803 ($1137- $3965) |

TB: Tuberculosis; ICER: Incremental Cost-Effectiveness Ratio; PLHIV: People Living with HIV; CLHIV: Children Living with HIV.

average DALY value of 2.44. When used in concurrent with LF LAM testing, the average cost rises to $298, while the average DALY value decreases to 1.93. The resulting incremental cost per DALY averted is $42, with a 95% UI of $18 to $345. Similarly, in the Philippines, LC-aNAAT on respiratory sample had average costs of $220 with average DALY value of 2.78, while concurrent use with LF LAM incurs an average cost of $238 and average DALY value of 2.13. The incremental cost per DALY averted is calculated to be $28 (95% UI: $12 to $249).

In Fig 4A and 4B, the ICER scatterplot based on 10,000 Monte Carlo simulations presents the cost-effectiveness results comparing the cost-effectiveness of LC-aNAATs using concurrent testing in respiratory sample and LF-LAM with LC-aNAATs stand alone on a respiratory sample among PLHIV in Malawi and the Philippines, respectively. Results consistently show incremental effectiveness greater than zero, indicating that concurrent testing methods utilizing LC-aNAAT result in improved diagnostic yield, and thus more DALYs averted compared to respiratory sample testing alone.

The analysis of CE acceptability curves, depicted in Fig 5A and 5B for Malawi and the Philippines respectively, showcases the proportion of simulations where the utilization of LC-aNAAT in concurrent testing among PLHIV is considered cost-effective compared to respiratory sample alone, across various WTP (willingness-to-pay) thresholds. In Malawi, at a WTP threshold of $50, concurrent testing is preferred in over 50% of simulations among PLHIV. Similarly, in the Philippines, at a WTP threshold of $35, concurrent testing is preferred in over 50% of simulations among PLHIV.

In Table 2 we observe the results of ICER per additional TB case diagnosed and ICER per death averted among PLHIV. The ICER per additional case diagnosed was $340 (95% UI: $272- $435) in Malawi and $290 (95% UI: $233-$373) in Philippines. Similarly, the ICER per death averted was $1,174 (95% UI: $504-$10,323) in Malawi and $949 (95% UI: $412-$8,329) in Philippines. In both Malawi and the Philippines, concurrent testing detected 63 additional TB cases per 1,000 PLHIV and averted 18 and 19 deaths, respectively, compared to the comparator (S5 Table).

## Concurrent use of LC-aNAAT on respiratory and stool sample and LF-LAM on urine among CLHIV with signs and symptoms of TB

The study findings presented in Table 1 shows the cost-effectiveness of concurrent use of LC-aNAAT on respiratory, stool sample and LF-LAM among CLHIV in Malawi and the Philippines. In Malawi, the average cost of

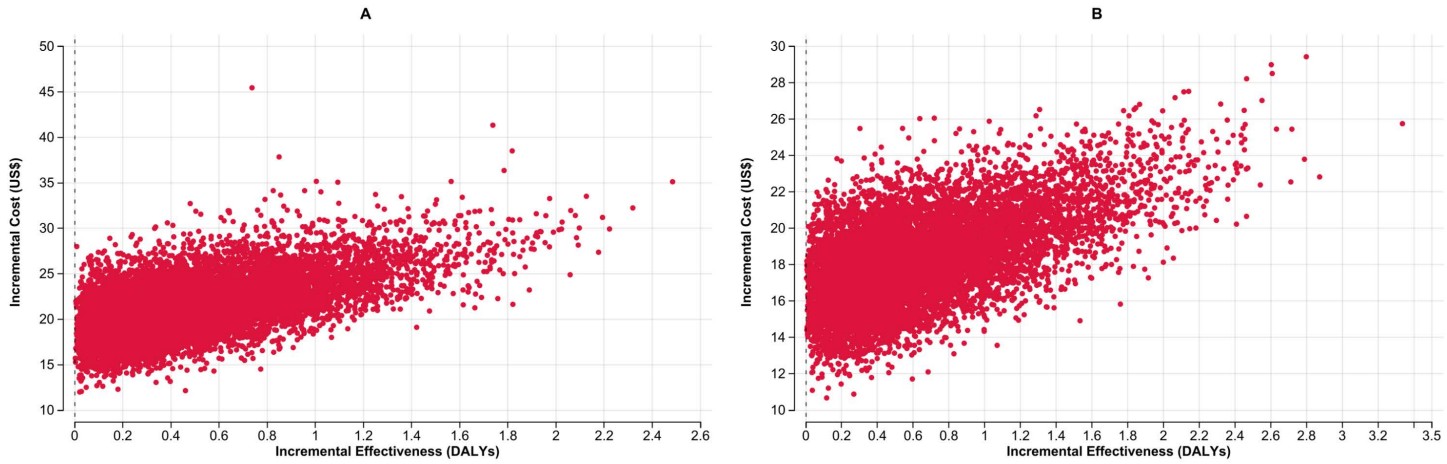

**Fig 4. Incremental cost-effectiveness scatter plots for the concurrent use of LC-aNAATs among PLHIV in Malawi (Panel A) and the Philippines (Panel B): Each scatterplot compares the ICER of concurrent testing versus respiratory sample testing alone, showing incremental cost (X-axis) against incremental effectiveness in DALYs averted (Y-axis).** Red dots represent 10,000 probabilistic simulations. LC-aNAAT: Low complexity automated nucleic acid amplification tests; DALY: Disability-Adjusted Life Year, ICER: Incremental Cost-Effectiveness Ratio; PLHIV: People Living with HIV; US$: United States Dollars.

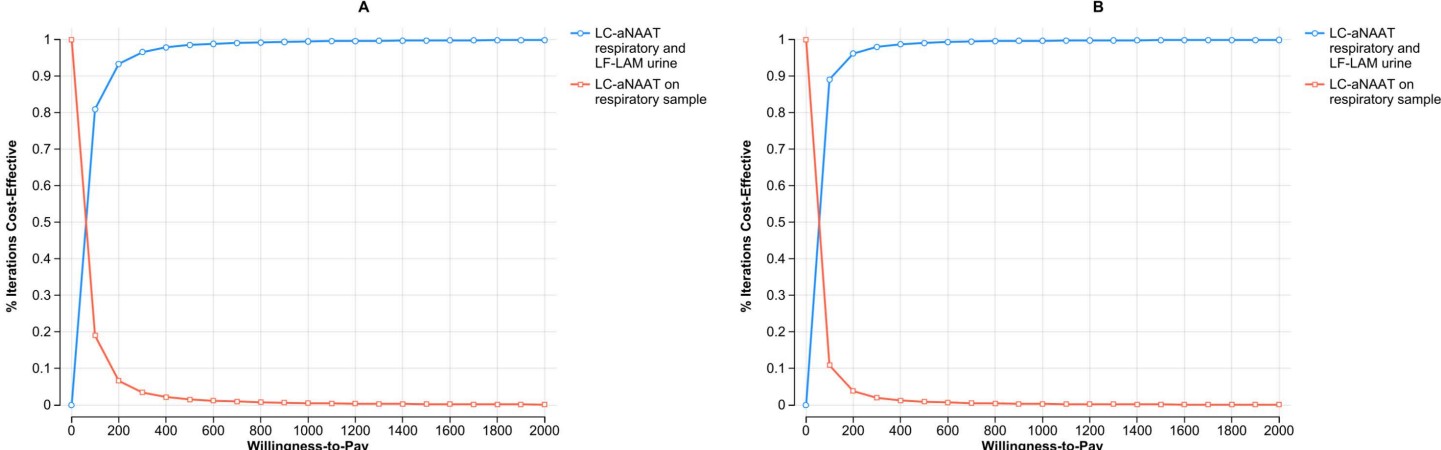

**Fig 5. Cost effectiveness acceptability curve on concurrent use of LC-aNAATs on PLHIV in Malawi (Panel A) and the Philippines (Panel B): CE acceptability curve: Represents the proportion of simulations where LC-aNAAT in concurrent testing is considered cost-effective compared to respiratory sample testing among PLHIV, at varying WTP thresholds.** Blue circles depict LC-aNAAT in concurrent testing; red circle LC-aNAAT in respiratory sample testing. LC-aNAAT: Low complexity automated nucleic acid amplification tests; CE: Cost Effectiveness; WTP: Willingness to Pay; PLHIV: People Living with HIV.

implementing L-aNAAT on respiratory samples is $319, with a corresponding average DALY value of 5.08. When used in concurrent, the average cost increased to $460, while the average DALY value decreased to 1.8. The resulting ICER per DALY averted is $43 (95% UI: $28 to $89). Similarly, in the Philippines, implementation of LC-aNAAT in respiratory sample alone costs $249 with average DALY value of 5.13, while concurrent use incurs an average cost of $345 and an average DALY value of 1.77. The ICER per DALY averted is calculated to be $29 (95% UI: $18 to $63).

In Fig 6A and 6B, the ICER scatterplot based on 10,000 Monte Carlo simulations, compares the cost-effectiveness of concurrent use of LC-aNAAT with use of LC-aNAAT on respiratory sample alone among CLHIV in Malawi and the Philippines, respectively. Results consistently show incremental effectiveness greater than zero, indicating that concurrent testing methods utilizing LC-aNAAT result in improved diagnostic yield, and thus more DALYs averted compared to respiratory sample testing alone.

The assessment of CE acceptability curves, illustrated in Fig 7A and 7B for Malawi and the Philippines respectively, reveals that in Malawi, at a WTP threshold of $44, concurrent testing is preferred in over 50% of simulations among CLHIV. Similarly, in the Philippines, at a WTP threshold of $30, concurrent testing is preferred in over 50% of simulations among CLHIV.

In Table 2, we observe the results of ICER per additional TB case diagnosed and ICER per death averted among CLHIV. The ICER per additional case diagnosed was $1,026 (95% UI: $870-$1,239) in Malawi and $701 (95% UI: $612-$842) in the Philippines. Similarly, the ICER per death averted was $2,425 (95% UI: $1,600- $4,911) in Malawi and $1,803 (95% UI: $1,137- $3,965) in the Philippines. Among CLHIV, concurrent testing identified 136 additional TB cases and averted 58 deaths per 1,000 children in Malawi, and 137 cases with 53 deaths averted in the Philippines compared to the comparator (S5 Table).

## Results of one-way sensitivity analysis

A one-way sensitivity analysis was conducted to identify the key variables exerting significant influence on the ICER. Variables with an influence exceeding 10% were selected for inclusion in the tornado diagram.

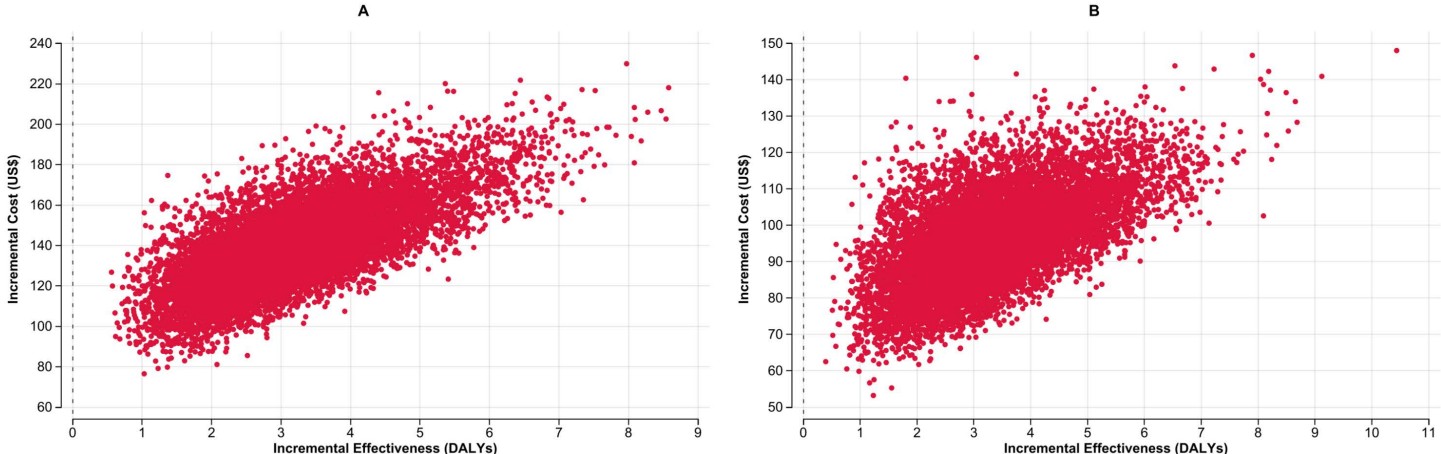

**Fig 6. Incremental cost-effectiveness scatter plots for the concurrent use of LC-aNAATs among CLHIV in Malawi (Panel A) and the Philippines (Panel B): Each scatterplot compares the ICER of concurrent testing versus respiratory sample testing alone, showing incremental cost (X-axis) against incremental effectiveness in DALYs averted (Y-axis).** Red dots represent 10,000 probabilistic simulations.LC-aNAAT: Low complexity automated nucleic acid amplification tests; DALY: Disability-Adjusted Life Year, ICER: Incremental Cost-Effectiveness Ratio; CLHIV: Children Living with HIV; US$: United States Dollars.

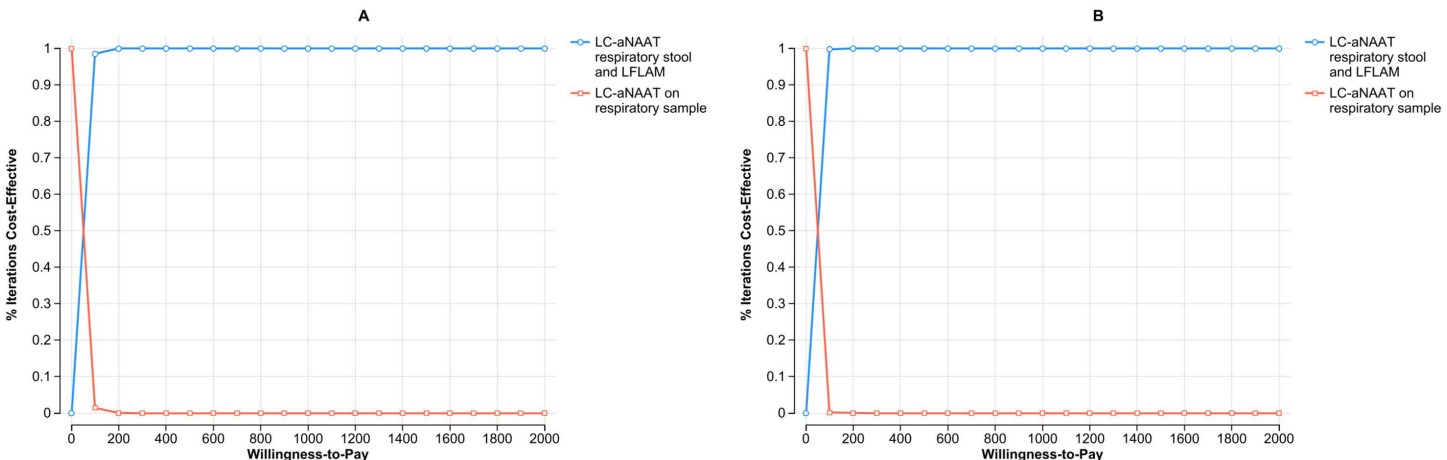

**Fig 7. Cost effectiveness acceptability curve on concurrent use of LC-aNAATs on CLHIV in Malawi (Panel A) and the Philippines (Panel B): CE acceptability curve: Represents the proportion of simulations where LC-aNAAT in concurrent testing is considered cost-effective compared to respiratory sample testing among CLHIV, at varying WTP thresholds.** Blue circles depict LC-aNAAT in concurrent testing; red circle LC-aNAAT in respiratory sample testing. LC-aNAAT: Low complexity automated nucleic acid amplification tests; CE: Cost Effectiveness; WTP: Willingness to Pay; CLHIV: Children Living with HIV.

S3 Fig display the results of a one-way sensitivity analysis represented as tornado diagrams, focusing on the concurrent use of LC-aNAAT among children in Malawi and the Philippines, respectively. Only those parameters that resulted in a change of +/-10% in the ICER are shown. The prevalence of TB among children with signs and symptoms of TB, the probability of providing a respiratory sample, and the sensitivity of LC-aNAAT in stool were the variables that resulted in the largest impact on cost-effectiveness results. The analysis reveals that as the prevalence of TB among children with signs and symptoms of TB increases, the ICER decreases, and vice versa. Similarly, an increase in the probability

of providing a respiratory sample leads to an increase in the ICER. These findings highlight the dynamic relationship between key variables and the cost-effectiveness of LC-aNAAT when used in concurrent testing among children, providing valuable insights for decision-making in healthcare resource allocation.

One-way sensitivity analysis was conducted on the concurrent use of LC-aNAAT among PLHIV, as shown in S4 Fig. The analysis revealed that the prevalence of TB among PLHIV with signs and symptoms of TB, the probability of providing a respiratory sample, and the per unit test cost of LF-LAM are the top three variables with the greatest impact on the ICER.

One-way sensitivity analysis was performed on the concurrent application of LC-aNAAT among CLHIV, as illustrated in S5 Fig. The analysis showed that the prevalence of TB among CLHIV with signs and symptoms of TB, the probability of providing a respiratory sample, and the specificity of LF-LAM are the top three variables exerting the most significant impact on the ICER.

### Results of scenario analysis

In our scenario analysis, we explored i) varying sputum production rates, ii) varying sensitivity of clinical diagnosis, and ii) using alternative comparators to calculate ICER per DALYs averted.

In scenarios where we adjusted the proportion of respiratory sample production rates (20%, 50%, and 80%), we observed an increase in the incremental cost per DALY averted as the rate of respiratory sample production increased among children, PLHIV and CLHIV in both countries. For instance, focusing on children in Malawi, when only 20% provide respiratory samples, the ICER for using concurrent LC-aNAAT is $250, which increases to $294 and $411 when 50% and 80% of children provide respiratory samples, respectively. The increase in ICER at higher respiratory sample production rates is due to greater DALYs averted in the comparator arm, as more individuals are diagnosed through respiratory testing. In contrast, the intervention arm sees less change, since alternate samples are already used when respiratory samples are unavailable, narrowing the DALY gap and raising the ICER. Further details of this scenario analysis are provided in S6 Fig for Malawi and the Philippines, respectively.

When we examine the scenario where the clinical diagnosis sensitivity ranges from 20% to 80%, we find that the ICER increases with an increment in the sensitivity of clinical diagnosis among all three target populations and in both countries, as demonstrated by S7 Fig This rise in ICER is driven by improved effectiveness in the comparator arm, as more TB cases are correctly identified clinically. Since the intervention already detects most cases through testing, the additional benefit is smaller—reducing the DALY difference and increasing the ICER.

In the base case, we considered LC-aNAAT used in respiratory samples as the comparator. However, in scenario analysis, we examined different comparators. For children and CLHIV, the comparator was LC-aNAAT in stool samples, while for PLHIV, it was LF-LAM in urine samples. In children and CLHIV, when the comparator is LC-aNAAT in stool sample alone, assuming all children provide stool samples, the ICER increases. In Malawi, when the comparator is the stool sample alone among children, the ICER increases from $253 to $912. This increment can also be observed in CLHIV, for both Malawi and the Philippines. This is because stool-only testing in the comparator arm offers better diagnostic coverage than respiratory-only testing, thereby narrowing the DALYs averted gap between the intervention and comparator arms. In contrast, for PLHIV, when the comparator is LF-LAM in urine sample alone, the ICER is significantly lower. This is due to the lower diagnostic accuracy of LF-LAM alone compared to respiratory testing, which increased the relative health gains of the intervention and led to a more favourable ICER. In Malawi, the ICER is $22, and in the Philippines, it is even lower at $9 as shown in Table 3.

### Discussion

Concurrent LC-aNAAT testing in children on respiratory and stool samples was cost-effective when compared to stand-alone test on respiratory sample with average incremental cost per DALY averted of $253 in Malawi and $156 in the

PLOS Global Public Health

**Table 3. Scenario analysis using different comparators.**

| Target group | Country | Comparator | ICER per DALYs averted |
|---|---|---|---|
| Children | Malawi | LC-aNAAT on respiratory sample | $253 |
| | | LC-aNAAT testing on stool sample alone | $912 |
| | Philippines | LC-aNAAT on respiratory sample | $156 |
| | | LC-aNAAT testing on stool sample alone | $430 |
| PLHIV | Malawi | LC-aNAAT on respiratory sample | $42 |
| | | LF-LAM testing on urnie sample alone | $22 |
| | Philippines | LC-aNAAT on respiratory sample | $28 |
| | | LF-LAM testing on urnie sample alone | $9 |
| CLHIV | Malawi | LC-aNAAT on respiratory sample | $43 |
| | | LC-aNAAT testing on stool sample alone | $153 |
| | Philippines | LC-aNAAT on respiratory sample | $29 |
| | | LC-aNAAT testing on stool sample alone | $113 |

LC_aNAAT: Low complexity automated nucleic acid amplification tests; ICER: Incremental Cost-Effectiveness Ratio; PLHIV: People Living with HIV; CLHIV: Children Living with HIV.

Philippines. Similarly, among PLHIV, concurrent testing with LC-aNAATs and urine sample LF-LAM was cost-effective with ICER per DALY of $ 42 in Malawi and $28 in the Philippines. Finally, among CLHIV, concurrent testing with LC-aNAAT on respiratory and stool sample alongside LF-LAM was also cost-effective, with ICER per DALY averted of $43 in Malawi and $29 in the Philippines.

This study demonstrates the cost-effectiveness of concurrent testing, even though they may lead to increased short-term expenditures for diagnostic services. While we did not conduct a formal Budget Impact Analysis (BIA) we acknowledge that implementing these strategies would require additional financial resources. Nonetheless, the potential long-term benefits, such as reducing the time to diagnosis, decreasing TB transmission [27,28], and contributing to the global goal of ending TB, justify the investment, especially in high-burden settings where early and accurate diagnosis can have substantial public health impacts. Our analysis included two emblematic settings—Malawi and the Philippines—and demonstrated cost-effectiveness in both. Importantly, these findings are generalizable to other settings, particularly those with similar TB and HIV prevalence, diagnostic challenges and resource constraints. However, it is worth noting that TB prevalence plays a significant role in cost-effectiveness outcomes. For example, Malawi's TB prevalence is lower than in many high-burden African and Asian countries [20], suggesting that the cost-effectiveness of concurrent testing could be even greater in settings with higher TB prevalence.

To our knowledge, this is the first economic analysis assessing the cost-effectiveness of concurrent use of LC-aNAAT to detect TB among children, PLHIV, and CLHIV across multiple specimen types. While some cost-effectiveness analyses exist on concurrent use among PLHIV, they primarily focus on Xpert MTB/Rif or Xpert Ultra, rather than evaluating the diagnostic accuracy of the entire class of LC-aNAAT. Nonetheless, the findings of those studies are consistent with our results. For example, a cost-effectiveness analysis by Fekadu et al. [29] on the concurrent use of Xpert Ultra and urine-based LAM found this to be a cost-effective strategy for diagnosing TB among PLHIV, with an ICER of $676.9 per DALY averted. Similarly, in a study by Esmail et al. and Reddy et al. [14,15] investigating concurrent testing with Xpert and LF-LAM among hospitalized PLHIV and found that concurrent use was likely the optimal diagnostic strategy. Although these studies were not conducted in identical TB/HIV prevalence settings as our study, the alignment in findings reflects a shared pattern that concurrent testing increases diagnostic yield and improves health outcomes. This consistency reinforces the robustness of our results and highlights the relevance of concurrent diagnostic approaches in diverse programmatic settings.

One-way sensitivity analysis identified key variables, such as TB prevalence and probability of providing respiratory samples, that influenced ICER across all three models. In addition, per unit test cost of LF-LAM, probability of death among those not initiated on treatment, and sensitivity of LF-LAM were also found to have an influence on the ICERs for PLHIV and CLHIV. As TB prevalence increased, ICERs decreased, highlighting the enhanced value of concurrent testing in high-prevalence settings. Conversely, higher probabilities of providing respiratory samples led to increased ICERs, underscoring the importance of non-respiratory sample testing in improving diagnostic yields. Scenario analyses further demonstrated variations in cost-effectiveness based on comparator strategies. For children and CLHIV, the comparator was LC-aNAAT in stool samples, while for PLHIV, it was LF-LAM in urine samples. Among children and CLHIV, the ICER increased when the comparator was LC-aNAAT in stool samples alone, assuming all children provided stool samples. For PLHIV, the ICER was significantly lower when compared to LF-LAM in urine samples alone. This difference is due to the nature of the comparators used. In the stool-based comparator for children and CLHIV, all individuals are assumed to provide stool samples, which improves diagnostic coverage and reduces the incremental benefit of the intervention—leading to a higher ICER. In contrast, LF-LAM alone has lower diagnostic accuracy than LC-aNAAT, resulting in greater health gains from the intervention and a lower ICER among PLHIV.

This study has limitations inherent to model-based analyses, including uncertainties in model structure and input parameters, though we used probabilistic sensitivity analysis (PSA) to address these uncertainties. Additionally, our model did not account for the impact of TB diagnosis on tuberculosis transmission. We found that concurrent testing with LC-aNAAT in children, PLHIV, and CLHIV is cost-effective compared to single respiratory sample testing. Including the effects of averted transmission would likely enhance health benefits and cost-effectiveness at the population level, as earlier and more accurate diagnosis reduces onward transmission, prevents secondary cases, lowers future treatment costs, and amplifies the overall benefit of the intervention. Some assumptions in our model contributed to its limitations. For instance, we assumed universal provision of non-respiratory samples for concurrent testing, which may have improved the cost-effectiveness results. We also assumed that all CLHIV and PLHIV are on ART without accounting for varying degrees of immunosuppression and its impact on diagnostic accuracy. This could have underestimated the cost-effectiveness, as concurrent testing might detect more TB cases in severely immunosuppressed individuals compared to respiratory sample testing alone.

## Conclusion

Concurrent use of LC-aNAAT emerged as a cost-effective strategy for diagnosing TB among children, PLHIV and CHLIV. Sensitivity analyses underscore the pivotal role of key factors, such as the likelihood of providing a respiratory sample, clinical diagnosis sensitivity and the prevalence of active TB in determining cost-effectiveness and should be carefully considered when contemplating implementing concurrent testing approaches across different settings. These findings suggest that concurrent use of LC-aNAAT can be a highly cost-effective diagnostic approach, especially in sub-Saharan Africa and Asia regions with high TB burden.

As we navigate the global fight against TB, these findings offer valuable guidance for healthcare decision-makers in optimizing diagnostic strategies and allocating resources effectively to combat this pervasive disease.

## Supporting information

**S1 Table. Epidemiological model parameters.**
(DOCX)

**S2 Table. Diagnostic accuracy model parameters.**
(DOCX)

**S3 Table. DALYs model parameter.**
(DOCX)

**S4 Table. Cost model parameters.**
(DOCX)

**S5 Table. Additional TB Cases Diagnosed and Deaths Averted per 1,000 Individuals by Target Group and Country.**
(DOCX)

**S1 Fig. Simplified model decision structure on the concurrent use of LC-aNAAT among PLHIV.**
(DOCX)

**S2 Fig. Simplified model decision structure on the concurrent use of LC-aNAAT among CLHIV.**
(DOCX)

**S3 Fig. One-way sensitivity analysis on the concurrent use of LC-aNAAT among children in Malawi and the Philippines.**
(DOCX)

**S4 Fig. One-way sensitivity analysis on the concurrent use of LC-aNAAT among PLHIV in Malawi and the Philippines.**
(DOCX)

**S5 Fig. One-way sensitivity analysis on the concurrent use of LC-aNAAT among CLHIV in Malawi and the Philippines.**
(DOCX)

**S6 Fig. Scenario Analysis – Varying respiratory sample production rate in Malawi and the Philippines.**
(DOCX)

**S7 Fig. Scenario Analysis – Varying clinical diagnosis sensitivity in Malawi and the Philippines.**
(DOCX)

**S1 Text. Supplemental Reference.**
(DOCX)

## Acknowledgments

The author gratefully acknowledges Ms. Ayushi Regmi for her invaluable assistance in formatting this manuscript.

## Author contributions

**Conceptualization:** Suvesh Kumar Shrestha, Pushpita Samina, Alexei Korobitsyn, Nazir Ismail, Alice Zwerling.

**Data curation:** Suvesh Kumar Shrestha.

**Formal analysis:** Suvesh Kumar Shrestha, Alice Zwerling.

**Funding acquisition:** Alice Zwerling.

**Investigation:** Suvesh Kumar Shrestha.

**Methodology:** Suvesh Kumar Shrestha.

**Supervision:** Alice Zwerling.

**Validation:** Suvesh Kumar Shrestha.

**Visualization:** Suvesh Kumar Shrestha, Alice Zwerling.

**Writing – original draft:** Suvesh Kumar Shrestha.

**Writing – review & editing:** Suvesh Kumar Shrestha, Funeka Bango, Pushpita Samina, Alexei Korobitsyn, Nazir Ismail, Alice Zwerling.

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
