## [Decision Letter · Decision Letter 0]

PGPH-D-25-00253

Concurrent use of low complexity automated NAATs for TB diagnosis and detection of resistance: a cost-effectiveness analysis

Dear Dr. Zwerling,

Thank you for submitting your manuscript to PLOS Global Public Health. After careful consideration, we feel that it has merit but does not fully meet PLOS Global Public Health’s publication criteria as it currently stands. Therefore, we invite you to submit a revised version of the manuscript that addresses the points raised during the review process.

We look forward to receiving your revised manuscript.

Kind regards,

Angela Devine, PhD

Academic Editor

Journal Requirements:

1. We have amended your Competing Interest statement to comply with journal style. We kindly ask that you double check the statement and let us know if anything is incorrect. 2. In the online submission form, you indicated that This study is a modeling analysis based on data extracted from publicly available sources, including previously published literature. All relevant data and parameter values used in the model are provided in the Supplementary Information files. Additional details can be made available upon reasonable request to the corresponding author. All PLOS journals now require all data underlying the findings described in their manuscript to be freely available to other researchers, either 1. In a public repository, 2. Within the manuscript itself, or 3. Uploaded as supplementary information. This policy applies to all data except where public deposition would breach compliance with the protocol approved by your research ethics board. If your data cannot be made publicly available for ethical or legal reasons (e.g., public availability would compromise patient privacy), please explain your reasons by return email and your exemption request will be escalated to the editor for approval. Your exemption request will be handled independently and will not hold up the peer review process, but will need to be resolved should your manuscript be accepted for publication. One of the Editorial team will then be in touch if there are any issues. 3. Please insert an Ethics Statement at the beginning of your Methods section, under a subheading 'Ethics Statement'. It must include: 1) The name(s) of the Institutional Review Board(s) or Ethics Committee(s) 2) The approval number(s), or a statement that approval was granted by the named board(s) 4. Please upload a copy of Figure 1, 2, 3, 4, 5, 6, 7  which you refer to in your text on page 7, 12, 13, 16, 17, 20, 21. Or, if the figure is no longer to be included as part of the submission please remove all reference to it within the text.

Additional Editor Comments (if provided):

Reviewers' comments:

Reviewer's Responses to Questions

**Comments to the Author**

1. Does this manuscript meet PLOS Global Public Health’s publication criteria ? Is the manuscript technically sound, and do the data support the conclusions? The manuscript must describe methodologically and ethically rigorous research with conclusions that are appropriately drawn based on the data presented.

Reviewer #1: Yes

Reviewer #2: Yes

Reviewer #3: Yes

2. Has the statistical analysis been performed appropriately and rigorously?

Reviewer #1: Yes

Reviewer #2: Yes

Reviewer #3: Yes

3. Have the authors made all data underlying the findings in their manuscript fully available (please refer to the Data Availability Statement at the start of the manuscript PDF file)?

Reviewer #1: Yes

Reviewer #2: Yes

Reviewer #3: No

4. Is the manuscript presented in an intelligible fashion and written in standard English?

Reviewer #1: Yes

Reviewer #2: Yes

Reviewer #3: Yes

5. Review Comments to the Author

Reviewer #1: Concurrent use of low complexity automated NAATs for TB diagnosis and detection of resistance: a cost-effectiveness analysis

Overview

Thank you for the opportunity to review this manuscript which evaluates cost-effectiveness of concurrent use of low complexity automated NAATs for TB diagnosis among children, PLHIV, and CLHIV. This is an important study that provides some insights into these important issues and can contribute towards global efforts to end TB.

Comments

Abstract

Consider replacing “measured” with “estimated or evaluated” in ‘”The analysis was conducted from a healthcare system perspective and measured...”

Methods

I think some brief description of what LC_aNAAT might look like in practice might help to put these findings in context. What tests are assumed to be used? Are these assumed to be universally available throughout all levels of care?

Results

I think it may be useful to include some more proximal outcomes in the results e.g cases detected and deaths. Although these are not the primary outcomes, I think they are important especially in programmatic settings. Equally important is including some measures of resource use e.g tests done and treatments required.

Figure 1: “The diamond symbol represents terminal node” - probably meant to say “triangle”.

Figures: Please define the abbreviations in all the figures.

Too many needless decimals are provided in the figures and in some cases making it difficult to read the actual values.

I think there are too many results presented in the main text and in my opinion it might be useful to focus on the main findings and a few SAs so that the message is not lost in the detail. The rest can be moved to the Appendix.

Discussion

The authors state that “these strategies may demand higher short-term expenditures…” - what is the impact of these strategies on the budget? Was a BIA done? I think it’s important to provide some reflection on what resources would be required and whether they would be affordable.

Reviewer #2: 1. On line 376, the table referred to is Supplementary Table 4, not Supplementary Table 3, as well as on line 437.

2. Please include the list of abbreviations in the supplementary tables and figures.

3. Please add the references used in the systematic reviews (SRs?) to Supplementary Table 2.

Reviewer #3: The authors evaluated the cost-effectiveness of the concurrent testing that includes low‐complexity automated nucleic acid amplification tests (LC-aNAAT) and antigen‐based lateral flow urine lipoarabinomannan (LF‐LAM) test among three groups of individuals, which were children under 10 years (HIV-negative only? It is not clear), HIV-positive individuals, and HIV-positive children under 10 years, in Malawi and the Philippines.

In the Abstract, it would be good to include that the study was conducted by using the data of Malawi and Philippines, before the results part. Also, in line 53, it would be better to provide the unabbreviated version of UR – the more common way to address this term is “uncertainty interval (UI)” for clarity.

The authors used P/CLHIV, PLHIV and CLHIV in different parts of the manuscript interchangeably. For instance, in line 41, it was referred as “… in children and people living with HIV (P/CLHIV)” and in line 109, the same definition was referred as “… among children and people living with HIV (PLHIV)”. For consistency, it would be useful to include the same abbreviation when indicating the same group of individuals throughout the manuscript. Also, it is not clear in the abstract that three different groups were considered in the study, which could be addressed as “children under 10 years”, “children with HIV, under 10 years”, and “adults with HIV”.

It is not clear why the PLHIV target group was not included in the statement regarding the gap in evidence, although it was indicated that existing studies on PLHIV did not include the concurrent testing of LC-aNAATs earlier in that paragraph (lines 132-139, page 5).

In the “Study Design” section, it would be helpful to briefly mention why Malawi and the Philippines were selected as the settings for this study, and also include the information of HIV prevalence among each target groups for both countries.

It would be beneficial for the audience to specify what was meant by PLHIV (Does that include only adults or does that also include adolescents?), also to explicitly state why they considered children only under 10 years old.

The statement in the “Intervention, and Comparator” section was repeated in the first paragraph of “Modeling” section by using different wording.

In Figure 1, LC-aNAAT was written as LC-aNNAT. Individual was written as “indivisual”. Also, the abbreviations “RS”, “RR”, “ATT” should be defined in addition to the other defined abbreviations in the footnotes of that Figure.

In the “Assumptions”, it would be better to state that at least these samples need to be provided by each group (lines 196-198) since it creates some ambiguity. (i.e., the text suggests that those are the only samples collected, which is not true, because these individuals could also provide respiratory samples.)

The parameters provided in the Table S1 (Supplementary Table 1) should be consistent. For instance, for the same group of individuals considered in this study, there were different definitions: “children under 5” and “under-five presumptive children”, “presumptive TB children” (The last two definitions should be changed to “children with presumptive TB”). Also, it seems that the name of the last target group should be “CLHIV”, not “children”.

In Table S2 (Supplementary Table 2), it would be better to cite the actual (published) sources for the parameter values rather than only referring to an ongoing (and I think it is unpublished) systematic review.

For the cost parameters (Table S3), my understanding is that values from another ongoing systematic review were used in the study. If so, please cite the references that include the cost values used in the Table S3 (instead of stating “SR” only).

In Line 229: Please change the part “The primary outcome being …” to “The primary outcome is …”.

In Line 238: The sentence that includes “… additional scenario analyses were performed.” is ambiguous since the main scenarios were not even identified in the manuscript yet (or the text before that sentence is missing).

Although DALYs were calculated in the study, the parameters table regarding the disability weights were not provided in the Supplementary tables (or in the main text). Please provide those parameters in the Supplement.

The average value of DALYs as a result of using the concurrent testing in the Philippines was given as 0.66 in the text, but it was stated as 0.62 in Table 1. The authors should confirm which number is correct and change one of those values accordingly.

On page 14, the authors provided the results of ICER per additional TB case diagnosed and ICER per death averted in the manuscript (lines 310-314), while the table that includes these results was given in the Supplement (as Table S4). It would be easier for the reader to follow, if the table was shown in the main text as well.

On page 18, in line 376, the results were provided in Table S4 (but the text says “in supplementary table 2”), and in line 387, the cost value was reported as $319, while that value was given as $320 in Table 3. Also, on page 22, in line 437, the results were provided in Table S4 (but the text says “in supplementary table 2”).

For Supplementary Figures 3a & 3b, it would be better if the x-axis tick marks, and the x-axis limits (not the ranges) were consistent.

For the scenario analyses where different respiratory sample production rates were considered, it would be good to add a rationale for choosing the rates as 20%, 50%, and 80%, and for choosing the sensitivity diagnosis ranges as 20%, 40%, and 80% (as shown in the Supplementary Figures 7a & 7b).

In the “Results of Scenario Analysis” section, it would be good to add a sentence to explain why there was a spike between the 50% respiratory sample production rate and the 80% respiratory sample production rate among children in both countries. Similarly, in the other scenario, a sentence to explain the same type of change (between 40% clinical diagnosis sensitivity and 80% clinical diagnosis sensitivity) is needed.

On page 24, where the authors describe the results of having different comparators, it would be beneficial to mention that the ICER refers to the ICER per DALY averted for clarity, since there were different ICER values in the manuscript. Also, it would be useful to add a few sentences to describe the changes in ICER values for different comparators. In other words, the reason for the increase in ICER values among children and CLHIV groups and the reason for the decrease in ICER values among PLHIV group, when a different comparator was used, need to be explained.

The table on page 24 that was provided as “Table 2” should be changed to “Table 4”.

Please cite the sources you considered for the statements in the paragraph on page 25 (lines 512-523).

In the paragraph (lines 525-535, on page 26), the authors stated that their results were consistent with the other studies, although it is not clear that other studies, which were cited in the manuscript, also focused on the cases in similar TB/HIV prevalence settings. Please mention the effects of the concurrent use of different testing strategies, and explain why the results of the other studies, despite evaluating a subset of diagnostic tests, aligned with this study.

In the paragraph where the authors discussed the results of one-way sensitivity analysis (lines 537-549, on pages 26-27), it would be useful to explain how using a different comparator for children and CLHIV than PLHIV affected the changes in the ICER values for the corresponding target populations.

Please explain in which ways including the effects of averted transmission in the modeling approach would increase the health benefits and cost-effectiveness at a population level (as stated in lines 555-557, on page 27).

In the “Conclusion” part, the clinical diagnosis sensitivity could be added to the key factor (as stated in lines 567-570, on page 28).

General comments: The authors need to check the manuscript thoroughly, and make sure that there are no grammatical errors or typos.

In the cost-effectiveness acceptability curves (Figures 3a & 3b, Figures 5a & 5b, and Figures 7a & 7b), it is not clear that the WTP threshold values, as stated in the text, actually correspond to more than 50% of simulations for each group and each country. Please indicate that information on the plots.

The Results section could have been reorganized without having sub-sections for each target group, since the current manuscript has repetitive statements for each target group (with the same outputs, but different values for each group of population considered in the study. As an example, lines 267-273, lines 333-339, and lines 397-402 have a similar text on the analysis of the same outputs for different target groups.). I believe it would be easier to follow to see the results based on outputs, which are given as stratified by target group. (For instance, Table 1, Table 2, and Table 3 could be merged.)

Rather than providing each Figure separately, the authors could create multi-panel figures that each panel shows the same output for one of the countries, stratified by the target group (For instance, Figures 3a and 3b (children) could be shown side-by-side. Figures 5a and 5b (PLHIV) and Figures 7a and 7b (CLHIV) would be the other panels in the same figure).

6. PLOS authors have the option to publish the peer review history of their article (what does this mean? ). If published, this will include your full peer review and any attached files.

**Do you want your identity to be public for this peer review?** For information about this choice, including consent withdrawal, please see our Privacy Policy .

Reviewer #1: **Yes: ** Nyasha Mafirakureva

Reviewer #2: No

Reviewer #3: No

---

## [Decision Letter · Decision Letter 1]

Concurrent use of low complexity automated NAATs for TB diagnosis and detection of resistance: a cost-effectiveness analysis

PGPH-D-25-00253R1

Dear Dr. Zwerling,

We are pleased to inform you that your manuscript 'Concurrent use of low complexity automated NAATs for TB diagnosis and detection of resistance: a cost-effectiveness analysis' has been provisionally accepted for publication in PLOS Global Public Health.

Best regards,

Angela Devine, PhD

Academic Editor

Reviewer Comments (if any, and for reference):

Reviewer's Responses to Questions

**Comments to the Author**

1. If the authors have adequately addressed your comments raised in a previous round of review and you feel that this manuscript is now acceptable for publication, you may indicate that here to bypass the “Comments to the Author” section, enter your conflict of interest statement in the “Confidential to Editor” section, and submit your "Accept" recommendation.

Reviewer #3: All comments have been addressed

2. Does this manuscript meet PLOS Global Public Health’s publication criteria ? Is the manuscript technically sound, and do the data support the conclusions? The manuscript must describe methodologically and ethically rigorous research with conclusions that are appropriately drawn based on the data presented.

Reviewer #3: Yes

3. Has the statistical analysis been performed appropriately and rigorously?

Reviewer #3: Yes

4. Have the authors made all data underlying the findings in their manuscript fully available (please refer to the Data Availability Statement at the start of the manuscript PDF file)?

Reviewer #3: Yes

5. Is the manuscript presented in an intelligible fashion and written in standard English?

Reviewer #3: Yes

6. Review Comments to the Author

Reviewer #3: (No Response)

7. PLOS authors have the option to publish the peer review history of their article (what does this mean? ). If published, this will include your full peer review and any attached files.

**Do you want your identity to be public for this peer review?** For information about this choice, including consent withdrawal, please see our Privacy Policy .

Reviewer #3: No
